

# Dissection of full-length transcriptome and metabolome of *Dichocarpum* (Ranunculaceae): implications in evolution of specialized metabolism of Ranunculales medicinal plants

Da-Cheng Hao[1,*], Pei Li[2,*], Pei-Gen Xiao[2] and Chun-Nian He[2]

[1] Dalian Jiaotong University, Dalian, China
[2] Chinese Academy of Medical Sciences, Beijing, China
[*] These authors contributed equally to this work.

## ABSTRACT

Several main families of Ranunculales are rich in alkaloids and other medicinal compounds; many species of these families are used in traditional and folk medicine. *Dichocarpum* is a representative medicinal genus of Ranunculaceae, but the genetic basis of its metabolic phenotype has not been investigated, which hinders its sustainable conservation and utilization. We use the third-generation high-throughput sequencing and metabolomic techniques to decipher the full-length transcriptomes and metabolomes of five *Dichocarpum* species endemic in China, and 71,598 non-redundant full-length transcripts were obtained, many of which are involved in defense, stress response and immunity, especially those participating in the biosynthesis of specialized metabolites such as benzylisoquinoline alkaloids (BIAs). Twenty-seven orthologs extracted from trancriptome datasets were concatenated to reconstruct the phylogenetic tree, which was verified by the clustering analysis based on the metabolomic profile and agreed with the Pearson correlation between gene expression patterns of *Dichocarpum* species. The phylogenomic analysis of phytometabolite biosynthesis genes, *e.g.,* (S)-norcoclaurine synthase, methyltransferases, cytochrome p450 monooxygenases, berberine bridge enzyme and (S)-tetrahydroprotoberberine oxidase, revealed the evolutionary trajectories leading to the chemodiversity, especially that of protoberberine type, aporphine type and bis-BIA abundant in *Dichocarpum* and related genera. The biosynthesis pathways of these BIAs are proposed based on full-length transcriptomes and metabolomes of *Dichocarpum*. Within Ranunculales, the gene duplications are common, and a unique whole genome duplication is possible in *Dichocarpum*. The extensive correlations between metabolite content and gene expression support the co-evolution of various genes essential for the production of different specialized metabolites. Our study provides insights into the transcriptomic and metabolomic landscapes of *Dichocarpum*, which will assist further studies on genomics and application of Ranunculales plants.

Corresponding authors
Da-Cheng Hao, haodc083@126.com
Chun-Nian He, cnhe@implad.ac.cn

## INTRODUCTION

The basal eudicot Ranunculales has seven families and more than 100 genera. Eupteleaceae is the earliest-diverging lineage, followed by Papaveraceae and core Ranunculales that includes five families (*Wang et al., 2009*). Circaeasteraceae and Lardizabalaceae are closer and basal to Menispermaceae (*Lian et al., 2020*; *Wang et al., 2020*), which is basal to Berberidaceae and Ranunculaceae (*Chen et al., 2020*). These families are rich in medicinal compounds (*Hao et al., 2015a*), especially alkaloids with versatile therapeutic activities. In Ranunculaceae, *Dichocarpum* Wang & Hsiao (*Xiao & Wang, 1964*) is a small genus of about 20 species mainly distributed from the warm to the subalpine zone of East Asia. The most recent common ancestor of *Dichocarpum* occurred in central China and Japan in the earliest Early Miocene (*Xiang et al., 2017*), suggesting an ancient vicariance event between China and Japan. Within mainland China, there might be three migrations at the species level that explain the expansion of *Dichocarpum* from central China to southeastern Yunnan, Hengduan mountains, and eastern Himalaya (*Xiang et al., 2017*). These migration events occurred in the Late Miocene to Early Pliocene, possibly influenced by the uplift of southeastern Qinghai-Tibetan Plateau (QTP) and the corresponding expansion of subtropical forests in China around that period. Around 13 species are endemic in China, which have specific distribution areas and occupy different ecological niches (Table S1). For example, *D. lobatipetalum* W.T. Wang & B. Liu and *D. malipoense* Tao grow in woods at elevations of 1,300 to 1,700 m (*Xie, Yuan & Yang, 2017*), preferring a limestone habitat. *D. auriculatum* (Franch.) WT Wang et PK Hsiao adapts to the wet land and mountain shade of 650 to 1,600 m and can be found under the sparse forest and/or beside the rock. These congeners evolve from a common ancestor and develop a variety of strategies (*e.g.*, specialized metabolites) to adapt to environmental stress during the long-term evolution, some of which are commonly used in traditional and folk medicine due to the therapeutic activities of phytometabolites.

The secondary (specialized) metabolism is essential in resisting natural enemies, defending pathogens, attracting pollinators and seed disseminators (*Zeng et al., 2021*), and mediating cooperation and competition between plants. As the defense mechanism of plants to adapt to the environment and the language to communicate with surrounding environment, many secondary metabolites are also economically important as they have health promoting effects and/or display therapeutic efficacy clinically (*Hao et al., 2015a*). As the strategy of adaptive evolution, different *Dichocarpum* species have overlapping but differential arsenals of secondary metabolites, which are used for their similar and different treatment spectra (Table S1). At least seven *Dichocarpum* species are traditional folk medicine in China, and their differential therapeutic efficacy is closely related with their respective phytochemical profile, especially the diversified specialized metabolites. In metabolomic analysis based on ultraperformance liquid chromatography–quadrupole time-of-flight–mass spectrometry (UPLC-Q-TOF-MS), 14 alkaloids, including five benzylisoquinoline alkaloids (BIAs), 16 flavonoids, cyanides, lactones, megastigmane glycosides, and organic acids were identified from *D. auriculatum* (*Li et al., 2019a*), implying that other congeners may also be rich in similar phytometabolites. However,

the phytochemistry of other related *Dichocarpum* species has not been reported, and the biosynthesis pathway of the above phytometabolites is totally unknown, as well as the genomic basis and evolutionary mechanisms of metabolic diversity.

The phylogenetically related species usually have similar chemical profiles (*Hao et al., 2015a*), implying that they have evolved similar biosynthetic pathways of some secondary metabolites to combat against various environmental stresses. Because of the limitation of read length of next generation sequencing (NGS), the full-length transcripts obtained by assembly are often not complete, while the third-generation sequencing technology represented by PacBio has effectively solved this problem (*Xu et al., 2015*). The PacBio platform uses single molecule real-time (SMRT) sequencing technology; with its advantage of super long sequencing reads, the quality full-length transcripts can be obtained without assembly. This technique has been used to obtain the full-length transcriptome or whole genome sequences of medicinal plants such as *Salvia miltiorrhiza* (*Xu et al., 2015*) and *Andrographis paniculata* (*Sun et al., 2019*), among others, and its advantages are just beginning to show in Ranunculales medicinal species (*Zhong et al., 2020*); it can greatly facilitate understanding the underlying connection between gene expression and phytometabolites. It is hypothesized that the full-length transcriptome, including the global expression of phytometabolite biosynthesis genes, of *Dichocarpum* can be elucidated without much difficulty by PacBio sequencing. The revealed genomic composition, along with phytometabolites identified by metabolomic analyses, represent an essential resource with versatile significance and utilities. For example, only by reconstructing the phylogenetic tree and inferring the evolutionary relationship as accurately as possible, can we comprehensively investigate the phylogeny and chemotaxonomy, and the multi-omics strategy enables the pharmaceutical resource discovery more efficient and facilitates the biodiversity conservation as well as developing ecological agriculture. In order to ascertain strategies of medicinal plants in adapting to environmental stresses and for the sake of guiding protection and ecological planting, the baseline data of the *Dichocarpum* transcriptome and metabolome are indispensable.

The omics studies of model plants inspire us from the perspective of comparative analysis. The unicellular green alga *Chlamydomonas reinhardtii* displays metabolic flexibility in response to a changing environment (*Strenkert et al., 2019*). Multiomics approaches, *e.g.*, transcriptome sequencing and metabolite profiling, showed their utility in resolving molecular events during a day in the life of *Chlamydomonas*. In *Chlamydomonas*, the role of individual metabolites, *e.g.*, carotenoids, phenolics, organic acids and vitamins, and non-coding RNAs that participate in defence during stress conditions can be understood by the combination of transcriptomics and metabolomics (*Kolackova et al., 2020*). The integration of metabolomics and transcriptomics can provide relatively precise information on gene-to-metabolite networks for identifying the function of unknown genes (*Tohge et al., 2005*). In a comprehensive analysis of metabolome and transcriptome of *Arabidopsis thaliana* over-expressing a MYB transcription factor, novel genes involved in flavonoid biosynthesis and novel anthocyanins were identified. Nevertheless, given the post-transcriptional modification and other levels of regulation, the dynamic response of the *Arabidopsis* metabolome to various factors may not be predicted by transcriptome changes (*Hildreth*

*et al., 2020*). The clues of regulatory mechanisms in Dichocarpum could also be identified by the integration of non-targeted metabolomics and full-length transcriptomics to gain a more objective understanding of the association between genes and specialized metabolites.

The major BIAs of *Dichocarpum* are protoberberine, aporphine and bisbenzylisoquinoline (*Li et al., 2019a*; *Li et al., 2019b*). They are important defense compounds against various biotic/abiotic stresses. Protoberberine type BIAs such as berberine, columbamine and coptisine (*Hagel et al., 2015*; *He et al., 2018*) have versatile pharmacological activities *in vitro* and in vivo. The bioactivities of aporphine BIAs and bis-BIAs are also confirmed (*Hao et al., 2015a*). The traditional and folk medicine knowledge suggest that *Dichocarpum* would be very useful for finding new pharmaceutical resources that are rich in BIAs and other medicinal compounds; species of this genus are the promising candidate of artificial cultivation in the form of wildlife tending and imitating wild habitat (*Huang & Chen, 2017*). The aim of the present study is to investigate the full-length transcriptomes of five Chinese endemic *Dichocarpum* species and their phytochemical profiles, especially the BIA diversity. These species are scrutinized from the perspective of pharmacophylogeny, and the association between gene expression profiles and the abundance of various phytometabolites is quantified for the first time. The phylogenetic reconstruction of a species tree and the gene trees further unravels the genetic diversity and the consequent metabolic diversity of *Dichocarpum*, and sheds light on the evolution and ecological adaptation of related Ranunculales plants.

## MATERIALS AND METHODS

### Plant materials

All *Dichocarpum* species were authenticated by one of the authors, Professor Pei-Gen Xiao. *D. auriculatum* (sample name SCEZ), *D. basilare* (SCJY), *D. malipoense* (YNMLP), *D. lobatipetalum* (YNLB), and *D. fargesii* (HBZL) samples were from Emei Mountain of Sichuan Province, Junlian County of Sichuan, Malipo County of Yunnan Province, Kunming of Yunnan, and Enshi of Hubei Province, respectively (Table S1). The specimen collections were approved by the Research Council of the Institute of Medicinal Plant Development, Chinese Academy of Medical Sciences (project number: 2018ZX09711001-008). As the whole plant of *Dichocarpum* is administered in folk medicine, three individuals were sampled for each species, and their roots, stems and leaves were mixed evenly for PacBio transcriptome sequencing. All voucher specimens are deposited in the Pharmacophylogeny Center at the Institute of Medicinal Plant Development of Chinese Academy of Medical Sciences, Beijing, China.

### PacBio sequencing of full-length transcriptome, Illumina sequencing and data processing

The RNAprep Pure total RNA Extraction Kit for polysaccharide/polyphenol rich plant (Tiangen, China) was used to extract total RNA from each *Dichocarpum* species. The isoform sequencing (Iso-Seq) library of each *Dichocarpum* species was prepared according to the standard protocol by using the Clontech SMARTer PCR cDNA Synthesis Kit and the BluePippin Size Selection System protocol as described by Pacific Biosciences (protocol PN

100-092-800-03). Three micrograms of total RNA were used for each library construction. The SMRT sequencing was conducted on the PacBio Sequel system.

The sequencing data were processed using the SMRT Link v7.0 software (https://www.pacb.com/support/software-downloads/). Circular consensus sequence (CCS) was generated from subread BAM files with the following parameters: min_length 50, max_drop_fraction 0.8, no_polish TRUE, min_zscore -9999.0, min_passes 1, min_predicted_accuracy 0.8, max_length 15000. CCS.BAM files were output, which were then classified into full length and non-full length reads using lima (https://github.com/pacificbiosciences/barcoding), and polyA was removed using refine software. Full-length FASTA files produced were then fed into the cluster step, in which the isoform-level clustering (n*log(n)) was performed, followed by final Arrow polishing with the following parameters: hq_quiver_min_accuracy 0.99, bin_by_primer false, bin_size_kb 1, qv_trim_5p 100, qv_trim_3p 30. The raw sequencing reads can be retrieved from China National GeneBank (CNGB) with the accession numbers CNX0280429-280435 (https://db.cngb.org/cnsa/project/CNP0001489/reviewlink/).

The NGS was performed on Illumina NovaSeq 6000 platform for quantifying *Dichocarpum* transcriptomes. The total RNA of 400 ng was used for sequencing library construction, and the sequencing strategy is PE150. Additional nucleotide sequencing errors in consensus reads were corrected using the Illumina RNA-Seq data *via* the software LoRDEC (*Salmela & Rivals, 2014*). Any redundancy in corrected consensus reads was removed by CD-HIT (-c 0.95 -T 6 -G 0 - aL 0.00 -aS 0.99) (*Fu et al., 2012*) to obtain final transcripts for the subsequent analysis. The raw Illumina sequencing reads can be retrieved from CNGB with the accession numbers CNX0280421-280425. The full-length sequences of 71,598 non-redundant transcripts (after clustering analysis) of five *Dichocarpum* species can be retrieved from CNGB with the accession numbers CNA0019224-19228.

## Bioinformatic and statistical analyses

The gene function was annotated as previously described (*Hao et al., 2015b*) based on the following databases: NR (NCBI non-redundant protein sequences), NT (NCBI non-redundant nucleotide sequences), Pfam (Protein family), KOG/COG (Clusters of Orthologous Groups of proteins), Swiss-Prot (manually annotated and reviewed protein sequence database), KO (KEGG Ortholog database), and GO (Gene Ontology). The software BLAST was used, with the e-value '1e−10' in NT database analysis; the software Diamond BLASTX was used, with the e-value '1e−10' in NR, KOG, Swiss-Prot and KEGG database analyses. The software Hmmscan was used in Pfam database analysis.

In protein coding sequence (CDS) prediction, the ANGEL pipeline (https://github.com/PacificBiosciences/ANGEL), a long-read implementation of ANGLE, was used to determine CDSs from cDNAs. The confident protein sequences of one species or its closely related species were used for ANGEL training, and then the ANGEL prediction was conducted for given sequences. The plant transcription factors (TFs) were predicted using iTAK software (*Zheng et al., 2016*). In long non-coding (lnc) RNA analysis, CNCI (Coding-Non-Coding-Index; https://github.com/www-bioinfo-org/CNCI), CPC (Coding Potential Calculator) (*Kang et al., 2017*), Pfam Scan (*Finn et al., 2016*), and PLEK

(*Li, Zhang & Zhou, 2014*) were used to predict the coding potential of transcripts. Transcripts with predicted coding potential by either/all of the above four tools were filtered out, and those without coding potential were the candidate set of lncRNAs. Simple sequence repeats (SSRs) of *Dichocarpum* transcriptome were identified using MISA (https://webblast.ipk-gatersleben.de/misa/), which allows the identification and localization of perfect microsatellites as well as compound microsatellites that are interrupted by a certain number of bases.

Based on PacBio and Illumina sequencing data, the gene expression levels were estimated by RSEM (*Li & Dewey, 2011*) for each sample: (1) Clean data of Illumina sequencing were mapped back onto the full length transcript sequences obtained from PacBio sequencing; (2) the read count for each transcript was obtained from the mapping results. Prior to differential gene expression analysis, for each sequenced library, the read counts were adjusted by the edgeR program package through one scaling normalized factor. Differential expression analysis of two samples was performed using the DEGseq R package (*Wang et al., 2010*). The $p$ value was adjusted to have the $q$ value as previously performed (*Hao et al., 2015b*). The $q$ value $< 0.005$ & $|log2(fold change)|> 1$ was set as the threshold for significantly differential expression. The GO enrichment analysis of differentially expressed genes (DEGs) was implemented by the GOseq R package (https://www.rdocumentation.org/packages/goseq/versions/1.24.0), in which gene length bias was corrected. GO terms with corrected $p$ value less than 0.05 were considered significantly enriched by DEGs. KOBAS software (http://kobas.cbi.pku.edu.cn/kobas3/?t=1) was used to test the statistical enrichment of DEGs in KEGG pathways.

In the full-length unique transcript model reconstruction, the non-redundant transcripts were processed with Coding GENome reconstruction Tool (Cogent v3.1, https://github.com/Magdoll/Cogent) with the parameters: –dun_use_partial. In Cogent, the input FASTA file was split into chunks of chunk_size to compute the K-mer profile. Each transcript family/cluster was further reconstructed into one or several unique transcript model(s) (referred to as UniTransModels) using a de Bruijn graph method. In alternative splicing (AS) analysis, error-corrected non-redundant transcripts (before Cogent reconstruction) were mapped to UniTransModels using GMAP-2017-06-20. Splicing junctions for transcripts mapped to the same UniTransModels were examined, and transcripts with the same splicing junctions were collapsed. Collapsed transcripts with different splicing junctions were identified as transcription isoforms of UniTransModels. AS events were detected with SUPPA (https://github.com/comprna/SUPPA) using default settings. In fusion gene analysis, due to the absence of *Dichocarpum* genome sequences, the reconstructed gene set was used as the reference genome, and UniTransModels are regarded as different genes. The principle of fusion gene detection is as follows: (1) A full-length transcript should be mapped to two or more gene loci in the reference genome; (2) each gene locus must be aligned with at least 10% region of this transcript; (3) the coverage ratio of the transcript to the reference genome must be more than 99%; (4) each aligned gene locus must be more than 100 Kb apart in the reference genome; (5) each gene locus must be supported by at least two short reads of NGS data. The parameter "–max-intronlength-ends 50000; -f 4; -z sense_force; -n 0" was used in GMAP software.

## Analysis of phylotranscriptomics and gene tree

According to BLAST results, Markov Cluster algorithm was implemented in the software OrthoMCL (https://sourceforge.net/projects/orthomcl/) to extract orthologs of five Dichocarpum species and four outgroup taxa, *i.e., Taxus mairei* (*Hao et al., 2011*), *Polygonum cuspidatum* (*Hao et al., 2012*), *Paeonia ostii* and *Paeonia lactiflora* (CNGB accession numbers CNX0350408 and CNX0350409 respectively), based on CDS predicted from their transcriptome datasets. PAML-CodeML (http://abacus.gene.ucl.ac.uk/software/#phylogenetic-analysis-by-maximum-likelihood-paml) was used to calculate Ka/Ks, and the ortholog with Ka/Ks > 1 was excluded in phylogenetic tree reconstruction. Muscle (https://www.ebi.ac.uk/Tools/msa/muscle/) was used to perform the translated amino acid sequence alignment; Gblocks 0.91b (http://molevol.cmima.csic.es/castresana/Gblocks.html) was used to optimize the results of protein sequence alignment, eliminate the poor alignment or sites that align to multiple regions, and the corresponding nucleic acid sequence alignment was generated. The third codon was excluded, and the individual aligned sequences were concatenated to generate a super-alignment file. PhyML 3.0 (http://www.atgc-montpellier.fr/phyml/) and maximum likelihood (ML) method were used to reconstruct the species tree.

In gene tree analyses, the phylogenetic relationships of members of representative gene families that contain functionally characterized genes involved in the biosynthesis of BIAs, flavonoids and terpenoids in *Dichocarpum* were elucidated. For each query sequence of *Dichocarpum*, the BLASTP search was conducted against the NR (https://blast.ncbi.nlm.nih.gov) and 1,000 Plant transcriptome database (https://db.cngb.org/blast/blast/blastp/). The sequences with an expected value lower than e-20 or within top 100 hits of the lowest expected values were retrieved, and the duplicated and partial sequences were removed. Functionally characterized genes were verified by checking the relevant publications. Muscle was used to perform protein sequence alignments. The best-scoring base substitution model and ML tree were inferred in MEGA X (*Kumar et al., 2018*), and the bootstrap analyses of 100 replicates were performed to obtain the support value of each branch.

The evolutionary relationship of 60 NMT amino acid sequences was analyzed by ML method. The initial tree(s) for the heuristic search were obtained automatically by applying NJ and BioNJ algorithms to a matrix of pairwise distances estimated using the JTT+F model, and then selecting the topology with superior log likelihood value. A discrete Gamma distribution was used to model evolutionary rate differences among sites (five categories, parameter = 1.4147). There were 429 positions in the final dataset. Sequences with ACxG, EC, MC, and PS prefixes are from the genome annotation datasets of *Aquilegia coerulea*, *Eschscholzia californica*, *Macleaya cordata*, and *Papaver somniferum*, respectively. The remaining sequences are from BLASTP search of NR database. In the evolutionary analysis of 37 BUP amino acid sequences by ML method, the evolutionary model is JTT+G (five categories, parameter 1.3639). There were 528 positions in the final dataset.

## Metabolomics

HPLC grade methanol and acetonitrile were purchased from Fisher Scientific (Pittsburgh, PA, USA). HPLC grade formic acid and analytical grade reagents were from Shanghai

Anpel Laboratory Technologies and Beijing Chemical Works, respectively. Water was produced with a Milli-Q Academic System (Millipore Corp., USA). The *Dichocarpum* metabolite extraction and high-resolution UPLC-Q-TOF-MS analysis were performed as previously described (*Li et al., 2019a*; *Li et al., 2019b*). The quality control (QC) sample was prepared by pooling aliquots of all extracts and was used to check MS shift over time and normalize data according to injection order. After MS system stabilization, samples were injected randomly, with one QC insertion every six samples. The plant extracts were analyzed in both positive and negative ion mode by UPLC-Q-TOF-MS. The positive mode was more convincing in the multivariate data analysis as it allowed the detection of a higher intensity response and more compounds. The continuum MSE raw data were imported into the Progenesis QI 2.3 software (Waters, Milford, MA, USA), and then subjected to noise reduction, peak detection/selection, and then alignment, which provide information of chromatographic retention time, m/z and intensity of characteristic peaks. Finally, the data matrix consisting of 4,630 features with m/z values, retention times, and intensities of putative compounds was exported as a CSV file in positive/negative ion mode. SIMCA software (14.0, Umetrics, Umeå, Sweden) was used in multivariate data analysis. The processed data was normalized with Pareto scaled prior to modelling. The in-house database was established to improve the confidence of compound annotation, which includes 275 published compounds of the related genera such as *Isopyrum*, *Aquilegia* (Table 1), *Paraquilegia*, *Semiaquilegia*, and *Thalictrum* of subfamily Thalictroideae. All compounds are saved in .mol format files and integrated into a .sdf format file through Progenesis SDF Studio (Waters, USA).

The differential retention time - exact mass pairs were imported back into Progenesis QI for compound identification. Three search methods were used. (1) Parameters of Progenesis MetaScope with an in-house library were: Both precursor tolerance and theoretical fragment tolerance 15 ppm. (2) Parameters of the commercial Metabolic Profiling CCS Library package (Waters, USA) were: Both precursor tolerance and theoretical fragment tolerance 15 ppm. Elements and corresponding number were set as H (0–50), C (0–50), N (0–5), and O (0–30), with precursor tolerance 15 ppm and isotope similarity 95%. 3) Standard compounds comparison.

For quantitative analysis of BIAs, the separation of multiple components was carried out using an ACQUITY UPLC H-Class liquid chromatography (Waters, USA) as previously described (*Li et al., 2019b*). The chromatographic separation was performed on a Waters cortecs C18 column (1.6 μm, 2.1 mm × 100 mm). The Waters Xevo TQD triple quadrupole tandem mass spectrometer (Waters, USA) equipped with an electrospray ionization (ESI) source was used for the mass analysis and detection. All data collected were analyzed and processed using the MassLynx (version 4.1, Waters, USA). The quantitative method validation was performed according to the standard of Editorial Committee of Chinese Pharmacopoeia (2020). To authenticate linearity, the standard solutions containing 12 reference compounds (Table 1) at a series of different concentrations were injected into the triple quadrupole tandem mass spectrometer and tested. The calibration curves of reference standards were constructed, and the limit of quantification (LOQ) and limit of detection (LOD) of 12 compounds were analyzed. The precision, reproducibility,

**Table 1  Absolute quantification of representative BIAs in *Dichocarpum* and related taxa.**

| Sample | SCEZ | SCJY | HBZL | YNMLP | *Enemion raddeanum* | *Aquilegia viridiflora* | *Thalictrum acutifolium* | *Rhizoma Coptidis* |
|---|---|---|---|---|---|---|---|---|
| Berberine (µg/g) | 3351.44 | 10.93 | 50.62 | 33.07 | 23.6 | 193.68 | 17.74 | 37450.8 |
| Jatrorrhizine (µg/g) | 1.47 | 0.48 | 18.29 | 23.92 | 1.38 | 20.19 | 0.72 | 1574.2 |
| Steponine (µg/g) | 440.69 | 0 | 2.05 | 6.72 | 0 | 24.95 | 1.01 | 0.75 |
| Magnoflorine (µg/g) | 1679.65 | 4202.6 | 456.16 | 2984.35 | 3369.1 | 1005.66 | 272 | 2773.81 |
| Menisdaurin (µg/g) | 3.43 | 0 | 0 | 820.91 | 3 | 0 | 0 | 0 |
| Phellodendrine (µg/g) | 1238.39 | 0 | 0 | 0 | 1504.54 | 0 | 0 | 0 |
| Columbamine (µg/g) | 103.45 | 1.19 | 18.64 | 24.25 | 3.24 | 18.49 | 1.16 | 1507.7 |
| Dauricine (µg/g) | 13.71 | 0 | 0 | 1575.62 | 0 | 0 | 0 | 0 |
| Palmatine (µg/g) | 1.44 | 1.81 | 1.56 | 1.78 | 1.3 | 1.32 | 2.65 | 6277.24 |
| Norcoclaurine (µg/g) | 54.02 | 31.67 | 9.1 | 47.39 | 2.51 | 6.27 | 96.94 | 1.74 |
| Tetrandrine (µg/g) | 18.23 | 1.06 | 2.9 | 1.92 | 356.61 | 6.6 | 28.98 | 1.04 |
| Corydaline (µg/g) | 1.51 | 2.16 | 2.32 | 2.24 | 2.07 | 2.31 | 1.16 | 2.44 |

**Notes.**

Menisdaurin is the sole non-BIA compound. Phylogenetically, *Enemion* and *Isopyrum* cluster together, which is closer to *Dichocarpum* than to *Aquilegia*; *Thalictrum* is closer to *Leptopyrum* and *Paraquilegia* than to the above four genera. All these genera belong to the subfamily Thalictroideae of Ranunculaceae.

stability and recovery were also evaluated according to the established method. The results of quantitative method validation were acceptable by comparing with the standards of Editorial Committee of Chinese Pharmacopoeia (2020). The biological replicates were analyzed in the absolute quantification of 11 BIAs and menisdaurin (Table 1), with at least three technical replicates for each sample.

Spearman rank/Pearson correlation coefficients were calculated to characterize the association between metabolome and transcriptome of *Dichocarpum*. The mean of all biological replicates of each species in the metabolome data and the expression value of each full-length transcript in the transcriptome data were calculated. The fold changes in each species were calculated in both the metabolome and transcriptome data and compared with the control species, *i.e.,* the five species were compared with each other. The coefficients were calculated from log2(fold change) of each metabolite and log2(fold change) of each transcript using the EXCEL program. Correlations corresponding to a coefficient with $p < 0.05$ were selected.

## RESULTS

### Processing of raw data of PacBio sequencing

The circular sequencing was performed on PacBio Sequel platform, and the quality sequencing read produced by SMRT sequencing process is called polymerase read. A total of 20.76 (YNLB, *D. lobatipetalum*)-22.96 (YNMLP, *D. malipoense*) Gb polymerase reads were obtained for five sequenced *Dichocarpum* species, corresponding to 315,190 (SCJY, *D. basilare*)-399,878 (SCEZ, *D. auriculatum*) polymerase reads. The original offline data were filtered, the adapters and the original data of < 50 bp length were removed to obtain subread sequences, from which the CCSs were generated without reference sequences. A total of 290,308–355,047 CCSs, with CNGB accession numbers CNX0350400–350404 (https://db.cngb.org/cnsa/project/CNP0001489/reviewlink/), were obtained for five *Dichocarpum* species (Datasets S1-S5); the maximum length of CCSs varied between 14,924 and 14,999 bp, from which 232,655–287,137 full-length non-chimeric (FLNC) reads, with CNGB accession numbers CNA0036084–36088, were generated (Datasets S6–S10). The FLNC sequences of the same transcript were clustered to get consensus sequence without redundancy, and the polished consensus sequences were obtained using Arrow software for subsequent analyses. A total of 25,323 (SCJY)-30,103 (HBZL, *D. fargesii*) polished consensus sequences, with CNGB accession numbers CNA0036089-36093, were obtained (Table S2, Datasets S11–S15); the N50 was 2,120 (SCEZ)-2,512 (YNMLP) bp. The PacBio sequences were further corrected by Illumina short sequencing reads to improve sequencing accuracy. According to 95% similarity between sequences, the corrected transcripts were clustered to remove redundant sequences, and 13,272–15,467 gene transcripts were generated for five species, which were combined to have the clustering analysis, then the unique and common transcripts among five samples were revealed (Fig. 1A). The genetically closer the two species are, the more transcripts they share. Interestingly, 3,205 transcripts were only shared by YNLB and YNMLP, while 2,994 were only found in both SCJY and SCEZ, which are reminiscent of their close phylogenetic relationship (*Xiang et al., 2017*; *Xie, Yuan & Yang, 2017*).

### Functional annotation and structural analysis of Dichocarpum genes

In five species, 12,584–14,807 genes were annotated by NR database. More than 95.3% of genes were annotated by at least one of seven databases. In GO annotation (Table S3), 4,468–5,298 genes of five species belong to "metabolic process", 4,322–5,128 have "catalytic activity", and 766–975 participate in the "response to stimulus". In KOG annotation, 307–437 genes are involved in "secondary metabolite biosynthesis, transport and catabolism", and 57–73 are involved in "defense mechanisms". In KEGG annotations, 45 *D. auriculatum* (SCEZ) genes are involved in metabolism of xenobiotics by cytochrome P450 (CYP), 39 non-redundant CYP transcripts participate in the xenobiotic biodegradation and metabolism, and 19 transcripts representing other enzymes are involved in this process. Transcripts participating in the degradation of pollutants, *e.g.*, chloroalkane and chloroalkene, naphthalene and other aromatic compounds, styrene, benzoate, and atrazine, were also identified. These imply that *Dichocarpum* could adapt well to the polluted environment. Many transcripts were involved in environmental adaptation, such as

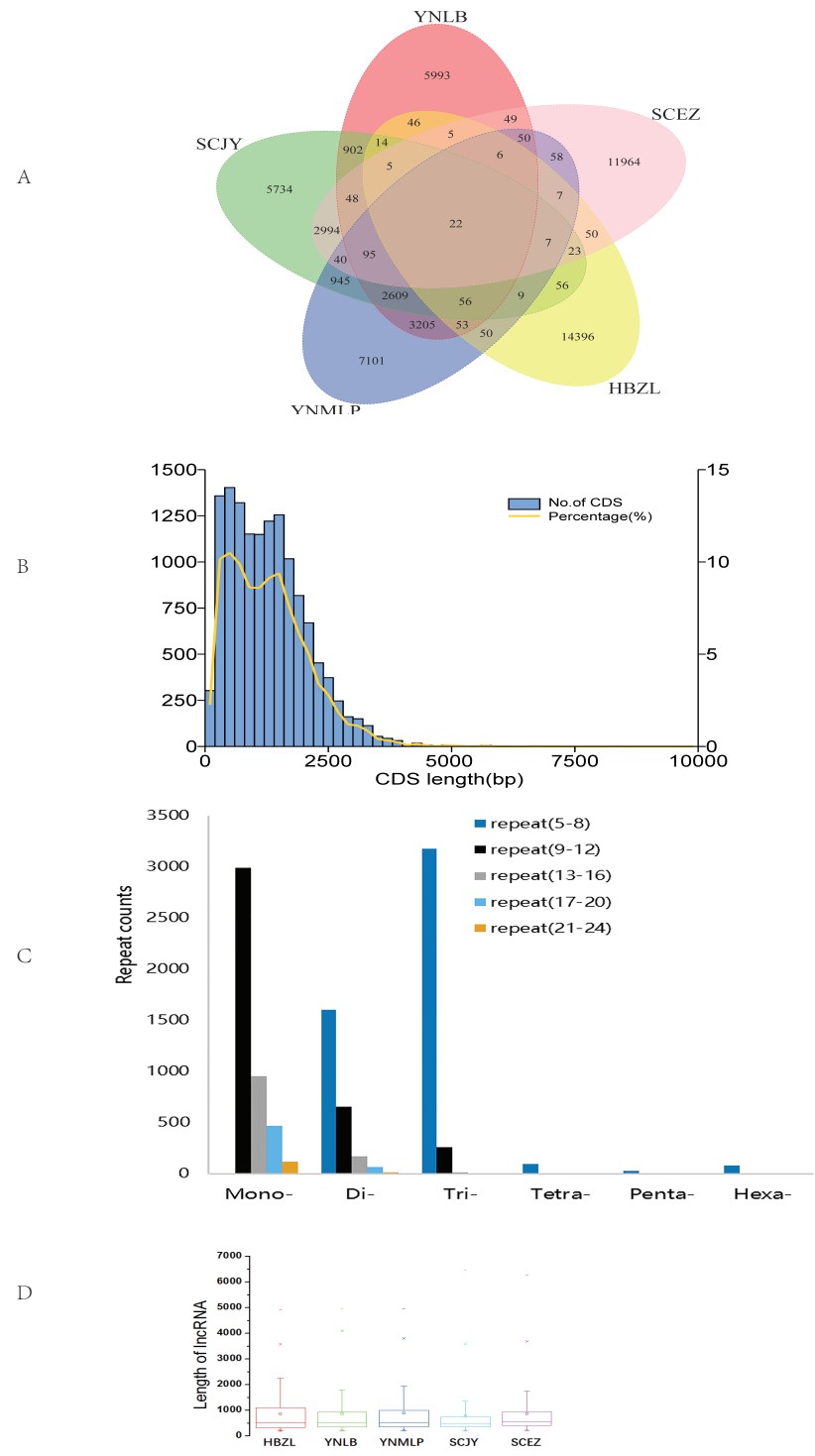

**Figure 1** **Data analyses of full-length *Dichocarpum* transcriptomes based on PacBio sequencing.** (A) Venn diagram showing the unique and common non-redundant transcripts among five *Dichocarpum* species. (B) CDS length distribution of *D. basilare* (SCJY). The other four species are similar. (C) SSR distribution of *D. auriculatum* (SCEZ). The other four species are similar. (continued on next page…)

**Figure 1 (…continued)**
(D) Box plot showing the length of lncRNA. Boxes represent 25–75% of the data, middle lines the median, and the small boxes inside represent the average values; the ends represent the minimum and maximum values (1.5 times lesser or greater than the lower or upper quantiles); outliers greater than 1.5 times and thrice the spacing of quartile are indicated by "×" and "−", respectively.

plant-pathogen interaction (153 genes) and circadian rhythm (36 genes); some transcripts participate in the environmental information processing, *e.g.*, Jak-STAT signaling pathway and TNF pathway. Many transcripts were homologous to the antimicrobial and anti-parasitic human genes, suggesting their roles in protecting *Dichocarpum* against various biotic stresses. Of note, numerous *Dichocarpum* genes could participate in the immune response, *e.g.*, NOD-like receptor signaling pathway (144 genes), Toll and Imd signaling pathway (114 genes), Toll-like receptor signaling pathway (102 genes), and the like. They confer resistance to pathogens and herbivores in the diverse living environments of *Dichocarpum*.

In CDS prediction, more than 90% of CDSs are between 250–2,500 bp (Fig. 1B). In TF prediction, the most common are WRKY, AP2/ERF-ERF (Table S4), and bZIP, followed by C2H2, C3H, and bHLH; GRAS, NAC, and MYB are also commonly identified. Many of these TFs are frequently involved in the regulation of plant defense against various biotic/abiotic stresses and biosynthesis of secondary metabolites (*Liu et al., 2021*). SSRs are widely and evenly distributed in plant genomes. The full-length transcriptome sequences generated by PacBio platform facilitate identifying much more SSRs as compared with NGS. In 7,788–9,467 SSR containing *Dichocarpum* transcripts, dinucleotide type SSR, mononucleotide SSR and trinucleotide SSR are predominant (Fig. 1C), and 1,063–1,348 compound type SSRs were identified in the full-length transcriptome of the respective species, which are difficult to detect by NGS. Except mononucleotide SSR, the most common repeat number of SSRs detected was 5–8. These markers are very useful in the evaluation of genetic diversity among congeners/populations for the preservation of endangered species and molecular breeding.

The high-quality plant genome/transcriptome assemblies and annotations provide novel research targets, including lncRNAs (Fig. 1D and 1S1A). PLEK predicted more lncRNAs (3,384–4,162) than Pfam (2,495–2,873), CPC (1,522–1,824) and CNCI (759–920) in five species (Fig. S1B). A total of 480-611 lncRNAs were simultaneously predicted by PLEK, CNCI, CPC and Pfam methods for five species (Fig. 1D). Eleven, 13, 17, 19 and 22 lncRNAs over 3,000 nt were identified from SCEZ, HBZL, SCJY, YNLB and YNMLP respectively. The mean length of these outliers was 3,594-3,966 nt, and the longest one was 6,446 nt. Based on annotations of seven commonly used databases, SCEZ_transcript1037/f3p0/3617 and SCJY_transcript1114/f2p0/3313 were similar to uncharacterized protein LOC109720399 of *Ananas comosus*. Except this, the similarity of lncRNAs between species was low, implying the species-specific epigenetic regulation. lncRNAs play important roles in plant development and stress responses (*Hou et al., 2019*).

In Cogent analysis, the number of transcripts contained in each of 39,787 reconstructed genes (UniTransModels) varies; those with one isoform transcript were the most (29,705;

Fig. S1C), followed by two (6,663), three (1,998) and four (713) isoforms. AS is an essential mechanism of gene regulation, and 21,163 isoforms, corresponding to 8,485 UniTransModels, were used to identify AS from five species. Among 1,037 AS events, Retained Intron (551) was predominant, followed by Alternative 3′ Splice Sites (253), Alternative 5′ Splice Sites (160) and Skipping Exon (59). Ten Alternative First Exons and four Alternative Last Exons were identified, and no Mutually Exclusive Exons was found. Totally 239 fusion genes were identified from five species when UniTransModels were used as the reference genome. All fusion genes consist of two blocks, which are from two different genes respectively; the average block 1 coverage varied between 46.2% (*D. fargesii*) and 87.8% (*D. malipoense*), and the length of fusion gene varied between 802 ± 428.5 bp (*D. auriculatum*) and 3,646.5 ± 3,878.4 bp (*D. lobatipetalum*). The above analyses for the first time reveal the structural complexity and diversity of *Dichocarpum* transcriptomes, which partially explain the functional diversity and high plasticity of *Dichocarpum* genomes in adapting to diverse and capricious environments.

## Gene expression of five congeners

In Illumina NGS, 41,428,838–47,681,210 clean reads were obtained for five species respectively, 82.61%–85.03% of which were mapped to the above-mentioned error-corrected/non-redundant full-length transcripts. The expression levels of most transcripts were lower than FPKM 5 (Fig. 2A), only 4.46–4.88% of 56,592 merged transcripts had the FPKM of > 60 (Table S6). SCJY had very different interval distribution pattern from those of other four species, which may be related to random sampling / random variation of samples; it does not affect the extraction of full-length gene sequences and mining of biosynthetic genes. The FPKM density distribution showed the largely similar gene expression pattern of each sample. In the correlation of gene expression pattern, the Pearson correlation between *D. malipoense* and *D. lobatipetalum* ($R^2$ 0.896, Fig. 2B and 2S1D) was much higher than that of other species pairs. The gene expression pattern of *D. fargesii* was least correlated with those of other species. In the hierarchical clustering analysis of 48,639 DEGs (Fig. S2), *D. malipoense* and *D. lobatipetalum* were closer than other species pair, and *D. fargesii*, again, was distinct. These results imply the kinship between five species.

In GO enrichment analysis of functional significance (Table S7), when compared with the genomic background, the GO functional items that were significantly enriched in DEGs were identified. The molecular function GO terms such as "transferase activity, transferring acyl groups", "hydrolase activity, hydrolyzing O-glycosyl compounds" and "nucleic acid binding" were significantly enriched in the specific species comparison, as well as the biological process GO terms such as "carbohydrate metabolic process" and "organonitrogen compound biosynthetic process". In KEGG enrichment analysis, pathways such as "Phenylpropanoid biosynthesis", "N-Glycan biosynthesis", "Starch and sucrose metabolism" and "Glutathione metabolism" were significantly enriched in the specific species comparison, which could partly explain the difference in phytometabolite profile and therapeutic efficacy among different *Dichocarpum* species.

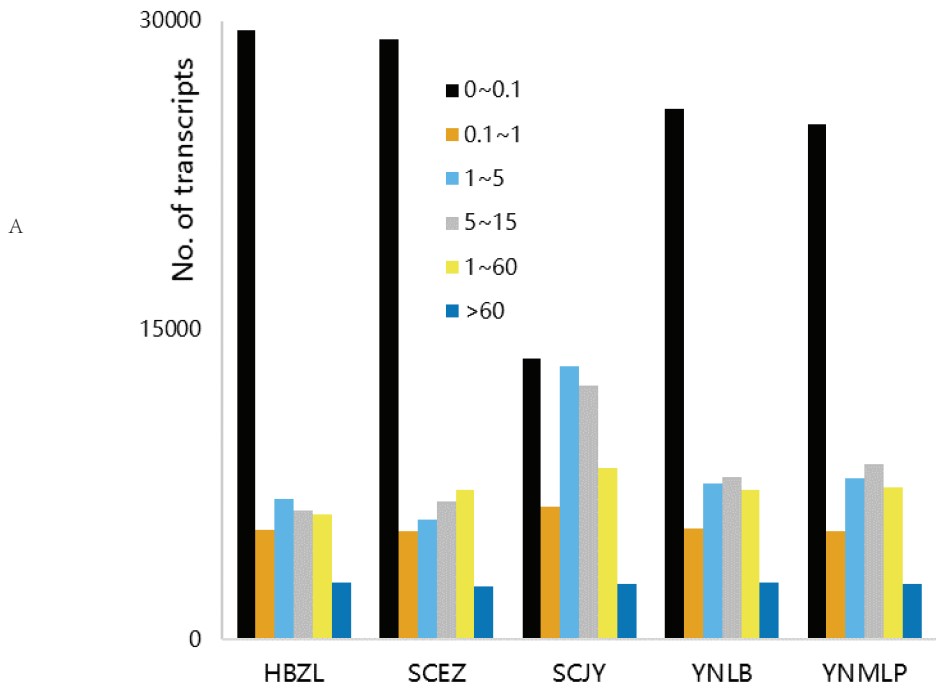

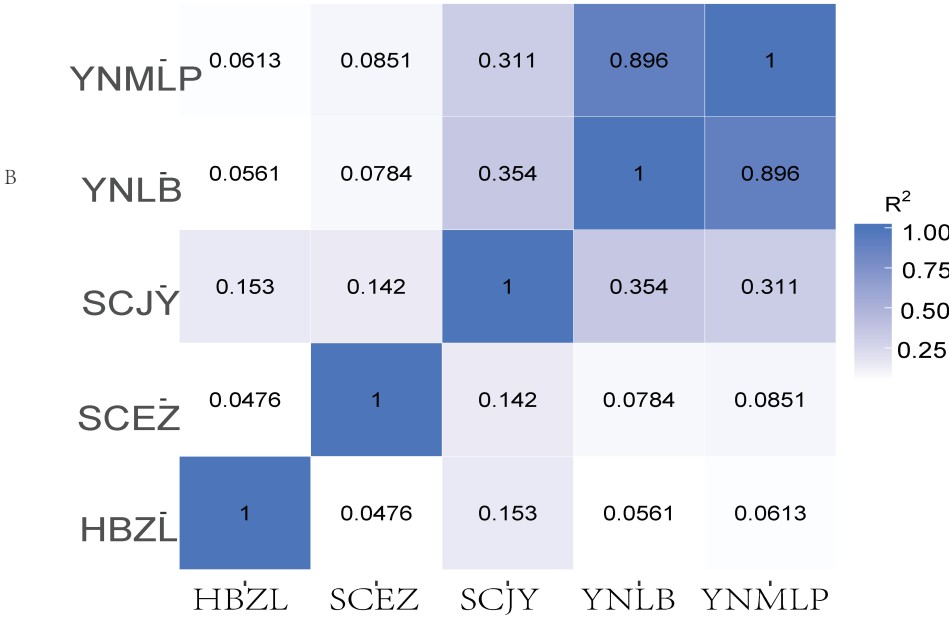

**Figure 2 Gene expression of *Dichocarpum*.** (A) Interval distribution of gene expression levels in five *Dichocarpum* species. (B) Pearson correlation between gene expression patterns of five species.

## Phylotranscriptomics and gene tree

Ninety-eight one-to-one orthologs of five *Dichocarpum* taxa were extracted from 19,610 ortholog groups by OrthoMCL, while 27 orthologs were obtained from nine taxa, including five *Dichocarpum* taxa and four outgroup species. All 27 evolutionarily conserved orthologs are under purifying selection ($K_a/K_s < 1$) and their 1st + 2nd codons were used in the species tree reconstruction (Fig. 3). *D. auriculatum* and *D. basilare* group together, while *D. malipoense* and *D. lobatipetalum* are very close; *D. fargesii* is basal to these four taxa. These results do not contradict with the correlation between their gene expression patterns (Fig. 2B and 2S2). In the ML tree inferred from three chloroplast (cp) markers and nuclear ITS (*Xiang et al., 2017*), *D. lobatipetalum* was basal to *D. fargesii*, possibly because not enough DNA markers were used, and cp DNA and nuclear ITS have special and distinct ways of evolution. (S)-norcoclaurine synthase (NCS), belonging to the Pathogenesis-related class (PR)10/Bet v1-like protein family (*Morris et al., 2021*), catalyzes the condensation of dopamine and 4-hydroxyphenylacetaldehyde to form (S)-norcoclaurine, which is the first step in the BIA biosynthesis. The NCS genes of some Ranunculales species have been functionally characterized (*Lee & Facchini, 2010*; *Marques et al., 2014*). At the root of NCS gene tree (Fig. S4A), SCEZ19213 and SCJY14783, possibly belonging to 2-oxoglutarate (OG)-Fe(II) oxygenase (2OGO) superfamily, were annotated as NCS-like transcripts. There are a few clades from Ranunculales families (Fig. S3A). SCJY19490 and YNLB20225 cluster with a functionally verified NCS AIT42265.1 of *Sinopodophyllum hexandrum* (Berberidaceae) (*Marques et al., 2014*). The functionally characterized NCS1 sequence from opium poppy is closer to orthologous sequences from *Eschscholzia californica* and *Corydalis linstowiana* of the same family. Similarly, some *Dichocarpum* sequences form a large group with two functionally known NCSs of *N. nucifera* (*Vimolmangkang et al., 2016*) (Fig. S3A). These suggest the monophyletic origin of NCS in the common ancestor of Ranunculales families.

The other four enzymes, *i.e.,* norcoclaurine 6-O-methyltransferase (6OMT), coclaurine-N- methyltransferase (CNMT), N-methylcoclaurine 3′-hydroxylase (NMCH, CYP80B subfamily), and 3-hydroxy-N-methylcoclaurine 4′-O-methyltransferase (4′OMT), are essential for the subsequent O-methylation, N-methylation, and hydroxylation steps in the biosynthesis of (S)-reticuline (*Hagel et al., 2015*; *Morris & Facchini, 2019*). The characterized genes encoding these enzymes of Ranunculaceae and Papaveraceae species are useful in inferring the phylogenetic positions and functions of *Dichocarpum* homologs. On the OMT tree, there are clades containing sequences exclusively from *Dichocarpum*, and there are also orthologous sequences from other Ranunculales species (Fig. S3B), which is similar to the NCS tree. The clade of 6OMT is closer to 4′OMT than to columbamine OMT (CoOMT), and three *Dichocarpum* OMTs (HBZL16802, 20954 and SCEZ18900) are basal to these clades and share more sequence identity with S-adenosyl-methionine (SAM)-dependent OMTs of *Camptotheca acuminata* (*Salim, Jones & DellaPenna, 2018*) and *Psychotria ipecacuanha* (*Nomura & Kutchan, 2010*). The latter two and *Vitis vinifera* OMTs 1-4 (*Guillaumie et al., 2013*) have dual substrates, *i.e.,* alkaloids and flavonoids. In SOMT clade, three *Dichocarpum* sequences (YNMLP19079, SCEZ18896 and SCJY15090) are closer to other Ranunculaceae SOMTs than to Papaveraceae homologs. Interestingly,
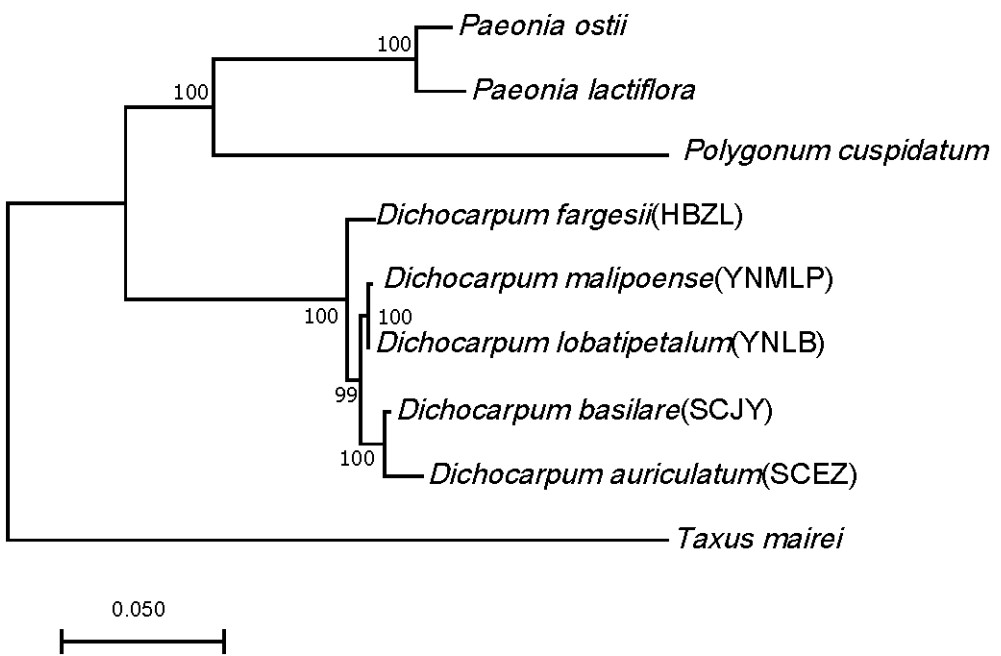

**Figure 3** **ML tree showing the phylogenetic relationship of five *Dichocarpum* taxa and four outgroup taxa, which is inferred based on 27 one-to-one single copy orthologs.** The model of nucleotides substitution is HKY85, and four classes are used in the discrete Gamma model. The bootstrap values are obtained using the default ''approximate Bayes branch supports'' in PhyML 3.0. The scale bar represents the number of base substitutions per site.

a flavonoid 3′OMT (SCJY15051) and 5′-desmethyl-yatein OMT (*Lau & Sattely, 2015*) are basal to the SOMT clade, implying the evolutionary linkage between flavonoid/lignan metabolism and alkaloid metabolism. These results are helpful to correct the misleading database annotation and suggest the unknown function of Ranunculales OMTs.

Analogously, on the phylogenetic tree of NMT (Fig. 4), eight *Dichocarpum* NMTs that are most likely authentic CNMTs were identified, and four are closer to pavine NMTs (*Liscombe et al., 2009*) than to reticuline NMTs (RNMTs). Interestingly, SCEZ15509 and 17045 cluster with CNMT of *Chlamydomonas reinhardtii* at the tree root. In the CYP80 tree (Fig. S3C), the CYP80B/NMCH clade contains a single Ranunculaceae subclade that groups with a Papaveraceae subclade. The above results suggest that all five genes responsible for the biosynthesis of (S)-reticuline are likely to be present in the Ranunculales common ancestor before the divergence of Ranunculaceae and Papaveraceae 110 MYA (*Li et al., 2020*).

The berberine bridge enzyme (BBE) and (S)-scoulerine 9-OMT (SOMT/9OMT) catalyze the first two steps in the biosynthesis of berberine from (S)-reticuline. (S)-tetrahydroprotoberberine oxidase (STOX) catalyzes the last step of synthesizing protoberberine type BIAs such as berberine, columbamine and coptisine (*Hagel et al., 2015*; *He et al., 2018*). BBE and STOX are FAD-linked oxidases (*Daniel et al., 2017*). The tree topology of these enzymes is similar to that of (S)-reticuline biosynthesis genes, suggesting their single-copy origin in the Ranunculales evolution (Fig. S3D); their

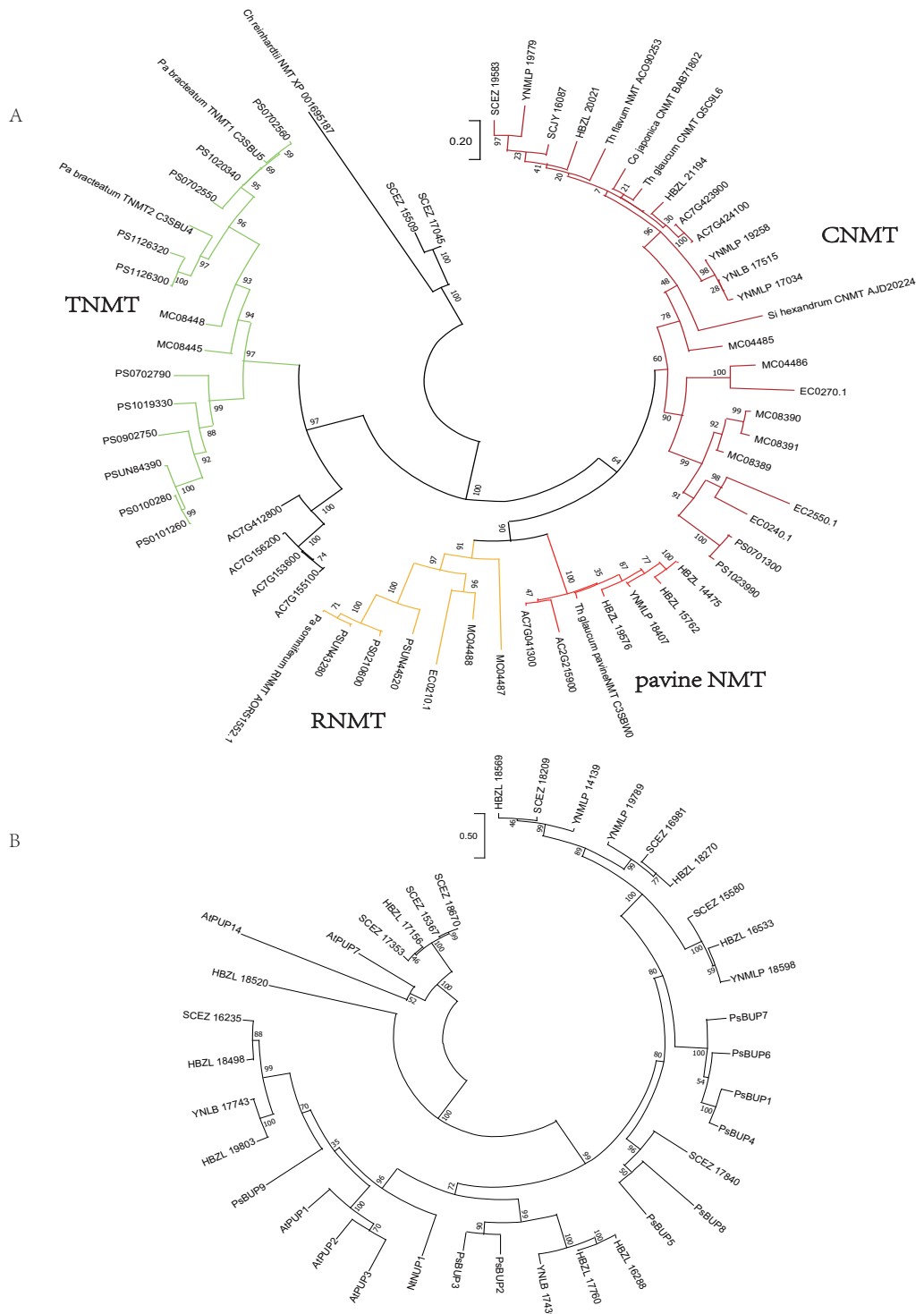

**Figure 4 Gene tree.** (A) Evolutionary analysis of 60 NMT amino acid sequences by ML method. The tree with the highest log likelihood (−15130.21) is shown. The percentage of trees in which the associated taxa clustered together is shown next to the branches. The tree is drawn to scale, with branch lengths measured in the number of substitutions per site. There were 429 positions in the final dataset. (continued on next page…)

**Figure 4 (…continued)**
Ch, *Chlamydomonas*; Co, *Coptis*; Pa, *Papaver*; Si, *Sinopodophyllum*; Th, *Thalictrum*. (B) Evolutionary analysis of 37 BUP amino acid sequences by ML method. The tree with the highest log likelihood (−16121.54) is shown. The bootstrap values are shown next to the branches. The scale bar represents the number of base substitutions per site. There were 528 positions in the final dataset. Ps, *Papaver somniferum*; At, *Arabidopsis thaliana*; Nt, *Nicotiana tabacum*. PUP, purine uptake permease; NUP, nicotine uptake permease.

orthologs are found in various Ranunculales genomes. Since protoberberine type BIAs are abundantly detected in *Dichocarpum* (detailed below), BBE should be in their genomes. Although no *Dichocarpum* sequence clustered with the *bona fide* BBE of *Coptis japonica* (BAM44344) (*Minami et al., 2008*), the authentic BBE could be identified by screening candidate transcripts suggested by the phylogenetic tree.

The purine permease type BIA transporters, BIA uptake permeases (BUPs), have been identified and characterized in opium poppy (*Dastmalchi et al., 2019*). Twenty-three full length *Dichocarpum* transcripts representing purine permease were identified in the SwissProt annotation and were combined with known BUPs in the phylogenetic analysis (Fig. 4B). Nine *Dichocarpum* sequences cluster with a *Papaver somniferum* BUP clade containing functional BUP1, 4 and 6, SCEZ17840 is closer to BUP5, while three other *Dichocarpum* sequences are closer to BUP2 and 3. These results provide important clues for the novel function of EamA-like transporter family.

### *Dichocarpum* chemodiversity revealed by untargeted metabolomics and its relationship with genetic diversity

The comprehensive metabolite profiling based on multiple analytical platforms contributes an improved landscape of overall metabolism occurring in taxonomically related BIA-producing plants. In metabolomic analysis, multiple types of primary and secondary metabolites were identified in five species (Table 1 and Table S8, Figs. 5 and 6A). The major BIAs of *Dichocarpum* are protoberberine BIAs, aporphine BIAs and bisbenzylisoquinoline alkaloids. The content of total alkaloids was the highest in *D. auriculatum* (peak area 610587.86; Fig. 6A), followed by *D. basilare*, *D. malipoense* and *D. lobatipetalum*, while *D. fargesii* had much less alkaloids (31446.34).

The content of individual alkaloid varied greatly in different species. For example, scoulerine was more abundant in *D. malipoense* (peak area 6698.2; Table S8) and *D. lobatipetalum* (3726.7) than in *D. basilare* (645.4) and *D. auriculatum* (327.2); magnoflorine was the highest in *D. basilare* (Table 1 and Table S8), followed by *D. malipoense*. Cheilanthifoline and menisperine were prominent in *D. basilare* and *D. auriculatum*; steponine was abundant in *D. auriculatum* (Table 1). Boldine and thalidezine were only abundant in *D. lobatipetalum* (Table S8). Compared with other congeners, the content of berberine in wild *D. auriculatum* was the highest, albeit it was only 8.9% of that of cultivated *Coptis chinensis*. The content of palmatine was comparable in five *Dichocarpum* species, so was that of thalsimine. The results of hierarchical clustering analysis based on the chemical profile agree with that of phylotranscriptomics.

Different pairs of alkaloids with absolute quantification were significantly correlated with a total of 66 genes (Spearman's rank correlation coefficient $r = 1$, $p < 0.001$; Fig. 6B,

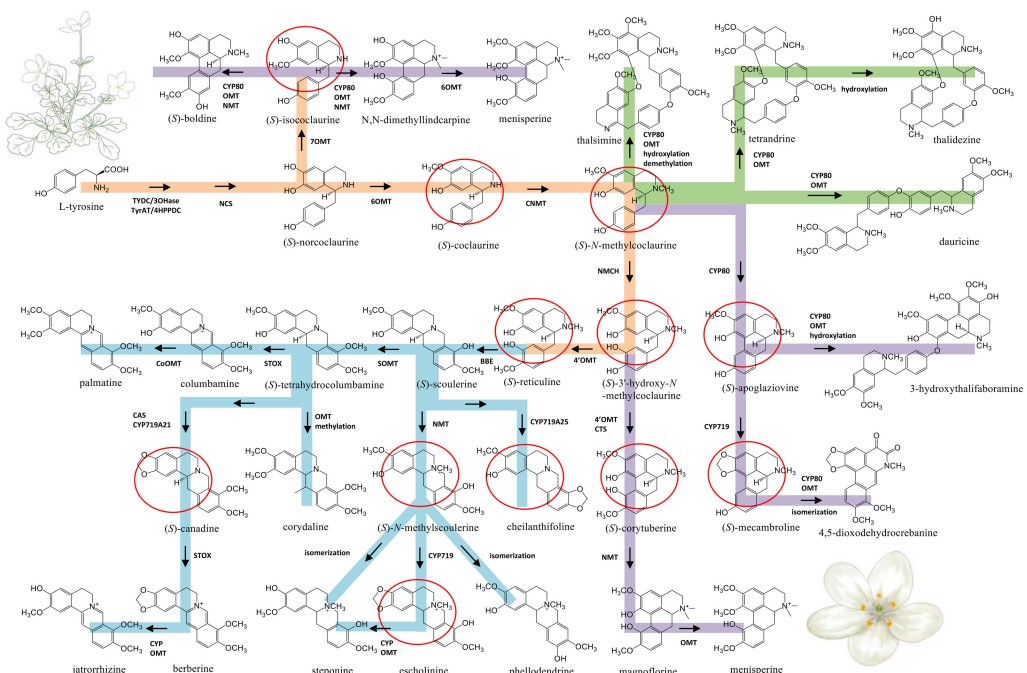

**Figure 5** **Proposed biosynthetic pathways of three major types of BIA identified from *Dichocarpum* species.** The common route of BIA biosynthesis is highlighted in orange, routes leading to protoberberine, aporphine and bis-BIA are shown in blue, purple and green, respectively. The major intermediates are marked by red circles. C-O and C-C coupling reactions could be catalyzed by CYP80 family members. Cleavage of the methylenedioxy bridge could be catalyzed by CYP *via* hydroxylation of the methylenedioxy carbon, followed by decomposition to the catechol metabolite and formic acid (*Richards et al., 2019*). The meaning of abbreviations is shown in Table S5.

Table 1 and Table S9A). Both berberine and tetrandrine were significantly correlated with 26 genes, *e.g.*, methyltransferases (MTs), solute carrier (SLC) family transporters and FAD dependent oxidoreductase, etc. Nineteen and 21 genes were substantially associated with steponine/columbamine and menisdaurin/dauricine respectively, which might be involved in stress/defense responses and multiple cellular processes. The homeobox domain and zinc finger containing TFs, myb-related and MADS-box TFs might be key to the regulation of BIA metabolism. These results were supported by the significant correlation between peak areas of BIAs and 88 genes (r = −0.892–0.976, *p* 0.004–0.049; Table S9B). Specifically, corytuberine synthase (CTS, CYP80G2) could be correlated with not only magnoflorine, but also other aporphine BIAs such as menisperine and 2,11-dihydroxy-1,10-dimethoxy-6,6-dimethyl-5,6,6a,7-tetrahydro-4H-dibenzo[de,g]quinolinium (N,N-dimethyllindcarpine, Fig. 5 and Table S8), as well as protoberberine BIA (cheilanthifoline).

## DISCUSSION

The Ranunculaceae genus *Dichocarpum* was established by *Xiao & Wang (1964)*; In China, many taxa of this genus are perennial herb in shady wet places on slopes or growing near rocks in forests of Sichuan, Yunnan, Fujian, and Hubei Provinces, etc. Species such as

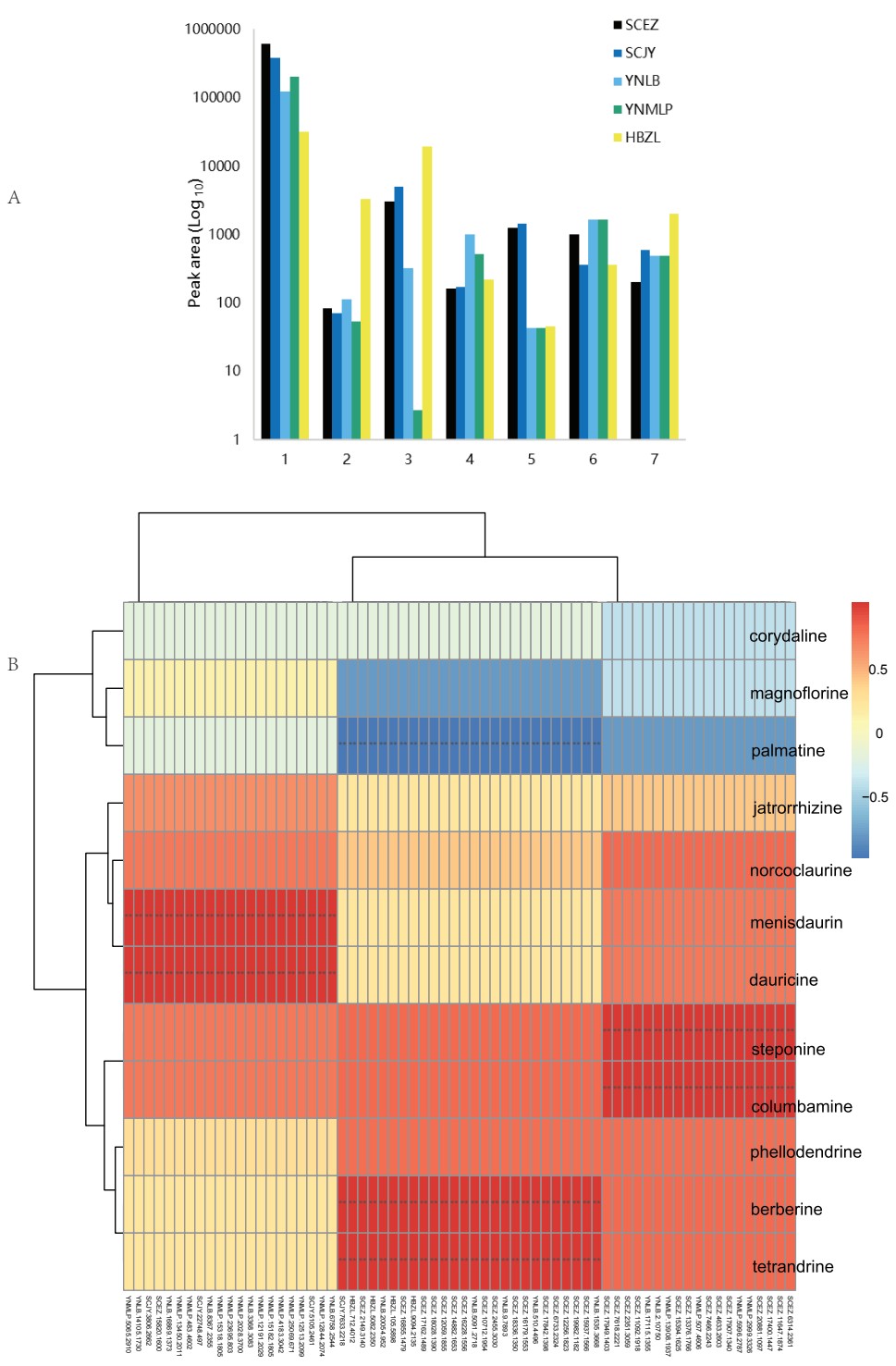

**Figure 6** *Dichocarpum* **phytometabolites and gene expression.** (A) Peak areas of *Dichocarpum* phytometabolites in LC-MS metabolomic analysis. 1, alkaloids; 2, triterpenoids and steroids; 3, flavonoids; 4, organic acids; 5, lactones; 6, saccharides; 7, others. (B) Correlation between BIA content of five *Dichocarpum* species and gene expression level. **, $p < 0.01$. Refer to Table S6A for details of the correlated metabolites and genes respectively.

*D. fargesii*, *D. auriculatum* and *D. basilare* are widely used traditional folk herb in southern China for clearing away heat and toxin, relieving cough, and eliminating phlegm (Table S1). The preliminary phytochemical studies show the abundant alkaloids in *Dichocarpum*, but their absolute content was not quantified, and other chemical constituents of the genus were much less known, let alone the biosynthesis of these phytometabolites. In the present study, the full-length transcriptomes of five representative *Dichocarpum* species are for the first time sequenced and characterized, the extensive gene duplication of this genus is revealed, which, along with the metabolite quantification, could help gain a deeper understanding of evolutionary trajectories and ecological/medicinal implications of *Dichocarpum* and related Ranunculales taxonomic groups.

## Phylogenetic relationship of *Dichocarpum* taxa

*Dichocarpum* is an East Asian endemic genus, and its phylogenetic relationship remains elusive. *Dichocarpum* was separated from *Isopyrum* and *Enemion* (Table 1) based on the features of leaves, follicles, petals, carpels, and seeds (*Xiao & Wang, 1964*), and contained 16 species belonging to two sections and two series. *D. fargesii* belongs to Ser. Fargesiana of Sect. Dichocarpum, while *D. auriculatum* and *D. basilare* belong to Ser. Sutchuenensia of the same section. *D. lobatipetalum* and *D. malipoense* (*Wang & Liu, 2015*; *Xie, Yuan & Yang, 2017*) are newly identified taxa with unclear phylogenetic position. An ML tree inferred from combined cp DNA and nuclear ITS data showed that *D. fargesii* is basal to *D. auriculatum* and *D. basilare* (*Xiang et al., 2017*), and *D. lobatipetalum* is basal to these three species. However, this is in contradiction with the morphological study, as *D. lobatipetalum*, like *D. auriculatum* and *D. basilare*, also belongs to Sect. Dichocarpum. Our phylotranscriptomic study based on 27 evolutionarily conserved orthologs supported the morphological grouping, which is also consistent with the ITS tree, clustering analysis of metabolomic data (not shown) and correlation/clustering analysis of gene expressions (Fig. 2B and 2S2). The gene expression patterns of species with closer genetic relationship are more possible to be closer, and vice versa. Different *Dichocarpum* species are subject to different ecological/environmental factors, which could increase the difference of gene expression. The cp markers *matK*, *trnL-F* and *trnH-psbA* have unique evolutionary characteristics (*Hao, Chen & Xiao, 2010*), and the incongruence between ITS tree and cp tree is not uncommon in Ranunculaceae (*Soza, Haworth & Di Stilio, 2013*). The phylogenetic analysis based on full-length ortholog sequences facilitates the elucidation of unclear relationship between closely related taxa.

## Polyploidy and gene duplication of *Dichocarpum*

Polyploids are often more climatically differentiated from their diploid parents than the diploids are from each other (*Baniaga et al., 2020*), and more adaptable to the shift of environmental factors. The discovery of extensive gene duplication and possible ancient whole genome duplication (WGD) in *Dichocarpum* lineages arouses the interest in polyploidy and its effects on the diversification of organisms' genotype and metabolic phenotype. There are many polyploid species in Ranunculales. For instance, there are quite some polyploid species in *Thalictrum* (*Soza, Haworth & Di Stilio, 2013*), and there is

intraspecific polyploidy. In *Circaeaster*, 2n = 30, x = 15, which is the paleopolyploid (*Nie, Sun & Gu, 2004*). In Qinghai-Tibet Plateau and Hengduan Mountains, neopolyploids were identified in 29 Ranunculaceae species (*Wang et al., 2017*), but paleopolyploid in this family is rare. For example, no paleopolyploid was found in *Aconitum*, *Anemone*, *Delphinium*, and *Ranunculus*, whereas nine, one, two and ten species of these genera are neopolyploids, respectively. In contrast, in *D. fargesii* (HBZL), *D. auriculatum* (SCEZ), and other species of Sect. Dichocarpum, 2n = 24, suggesting the paleotetraploid (*Fu, 1988*; *Yang et al., 1993*). In addition, species of Sect. Hutchinsonia are paleohexaploids. These data, along with multi-copy of metabolic genes and numerous paralogs revealed by the present study (Fig. 4 and Fig. S3; see below), raise concerns of whether there is a unique ancient WGD in *Dichocarpum*, which warrants further quantitative study.

The BIA metabolites such as (S)-reticuline, berberine, and magnoflorine are commonly found in Ranunculales plants, which suggest shared evolutionary history of their biosynthesis pathways. Here the gene tree analysis could help resolve key evolutionary steps with regard to new BIA pathways in these plants. The current biochemical and molecular insights of BIA biosynthesis are mainly from Ranunculales species, especially *Papaver somniferum*, *Eschscholzia californica*, and *Coptis* species (*Lee & Facchini, 2010*; *Li et al., 2020*; *Liu et al., 2021*). Some genes involved in the biosynthesis of BIAs have been functionally characterized, which, in conjunction with full-length *Dichocarpum* transcripts of this study and other homologous sequences in NCBI NR and CNGB 1KP databases, were amenable in gene tree analyses of BIA pathway. The gene search and phylogenetic tree reconstruction shed light on the timing of gene duplications and neofunctionalization/subfunctionalization events in the context of BIA biosynthesis.

## Evolutionary trajectory of *Dichocarpum* biosynthesis genes

In gene tree analysis, it should be noted that NCS-like sequences should be used along with functionally characterized NCS genes, so as to better infer the potential utility of NCS-like sequences. Different *Dichocarpum* sequences cluster with different *bona fide* NCSs (Fig. S3A), including those of basal eudicot sacred lotus (Proteales); sequences of the same plant family/genus do not cluster together. These support the ancient origin of NCS activity, which may not evolve independently in orders Ranunculales and Proteales. Yet, the function of NCS candidates should be fully characterized in the future study. Of note, no orthologs of canadine synthase (CAS) and columbamine OMT (CoOMT) genes involved in protoberberine biosynthesis were found in our *Dichocarpum* transcriptomes, which does not mean that there are no such genes in *Dichocarpum* species, as abundant protoberberine type BIAs, *e.g.*, cheilanthifoline, steponine, palmatine, and berberine, were detected (Table 1 and Table S8, Fig. 5). Similarly, the abundance of aporphine BIAs, *e.g.*, magnoflorine, boldine, and menisperine, in *Dichocarpum* species examined suggests the presence of all orthologs involved in aporphine biosynthesis, albeit the absence of RNMT (Fig. 4A) in our *Dichocarpum* transcriptome datasets. These suggest the complementarity of two omics methods. Without the aid of metabolomics, our understanding of the mechanisms, regulation and evolution of the biosynthesis of Ranunculales specialized metabolites cannot be complete. Within Ranunculaceae, the distribution of BIA, triterpene

and other metabolites among species is very uneven (Table 1 and Table S8), which cannot be solely explained by the different expression level of biosynthetic genes in the respective species, as the regulation mechanisms of defense arsenal are complex against versatile environmental conditions, *e.g.*, post-transcriptional regulation, translational/post-translational modulation, and epignomic modification, and such like.

The gene tree topologies propose that the protoberberine/aporphine/bisbenzyliso-quinoline biosynthetic pathways in Ranunculales evolved prior to the divergence of Ranunculaceae and Papaveraceae at 110 MYA (*Li et al., 2020*). The extensive distribution of berberine and magnoflorine in species of most Ranunculales families is consistent with this proposal (*Hao et al., 2015a*). Species with the similar metabolomic profile are more possible to possess the similar bioactivity profile (*Gonulalan et al., 2020*), therefore, bioprospecting medicinal taxa in Ranunculales is promising and this plant order represents a hot clade in drug discovery and development.

Plant specialized metabolites are the largest and most diverse pool of natural products, which play the essential role in biotic/abiotic stress response, UV defense, and disease resistance. The phytometabolites such as alkaloids, phenylpropanoids/flavonoids and terpenoids are crucial to a species' viability. Our tree analyses propose a monophyletic origin for all genes required for the biosynthesis of BIA, flavonoid/anthocyanin, and other specialized metabolites (*e.g.*, triterpenoid) in the order Ranunculales. The biosynthetic pathway leading to the production of protoberberine, aporphine and bisbenzylisoquinoline alkaloids arose before the divergence of Papaveraceae, Berberidaceae and Ranunculaceae families between 110 and 122 MYA (*Li et al., 2020*), probably so did those leading to the generation of other major defense phytometabolites. On the other hand, the BIA biosynthesis, as well as biosynthesis of other phytometabolites, in *N. nucifera* (Proteales, another early-diverging eudicot order; *Sun et al., 2016*), is less likely to have the parallel evolution, as relevant genes share common ancestors with those of Ranunculales (Fig. S3), implying the very ancient origin of specialized metabolite biosynthesis. The split of five *Dichocarpum* species was inferred to be 17.28 MYA (*Xiang et al., 2017*). With this reference, we calculated the number of synonymous substitutions per synonymous site ($K_s$) between paralogous gene pairs to estimate the timeline of gene duplication. The average and median of $K_s$ of biosynthetic genes are 0.1702–0.2952 and 0.1307–0.3251, respectively (Table S10). Based on a synonymous substitution rate (r) of 6.98 per billion years (*Guo et al., 2018*) and $T = K_s/2r$, the time (T) for the duplication would be in the range of 12.2–21.1 (average) and 9.36–23.3 (median) MYA, respectively. Thus, the segmental duplication and/or WGD event of *Dichocarpum* might occur in Miocene, when there was the rapid rise of East Asian subtropical evergreen broadleaved forests, and sympatric Berberidaceae/Lardizabalaceae underwent the biogeographic diversification (*Chen et al., 2020*; *Wang et al., 2020*). The gene families involved in the BIA biosynthesis and transport went through recent rapid expansion in *Dichocarpum* and probably related genera (Fig. 4 and Figs. S3A–S3D), making these genera a treasure house of ecological/medicinal resources.

## CONCLUSIONS

In summary, this study demonstrates the effectiveness of high throughput full-length transcriptome sequencing, species and gene tree analyses, and gene/metabolite correlation analysis in elaborating gene expression and metabolic profiles, unearthing the hidden connections between transcriptome and metabolome, and clarifying key evolutionary facts with respect to the plants' adaptibility in the capricious environments. Polyploidy is widespread across the plant Tree of Life (*Román-Palacios, Molina-Henao & Barker, 2020*), which could play a major role in driving the long-term evolution of Ranunculaceae and related taxonomic groups and shaping present-day diversity patterns across Ranunculales. In addition to the structural genes of biosynthetic pathways, the associated TFs probably co-evolve to achieve the coordinate expression of biosynthetic genes. In the future work, the evolution of specialized metabolite biosynthesis should be investigated within the context of whole genome sequencing of multiple related species, and genes not covered by the transcriptome sequencing could be mined further. The inferred BIA biosynthetic pathways based on both transcriptome and metabolome datasets would help identify more genes leading to chemodiversity of alkaloids. It will be interesting to see whether the metabolic gene clusters are present in *Dichocarpum* and other Ranunculaceae genera. Such a cluster should include both structural genes and regulatory genes. In addition, the gene regulation at genome organization levels should also be taken into account.

## ACKNOWLEDGEMENTS

We thank the Section Editor, the Academic Editor and two anonymous review experts for their constructive comments and suggestions.

### Funding

This research is supported by the Major Scientific and Technological Special Project for "The Drug Innovation Major Project" (No. 2018ZX09711001-008), and the Innovation Team and Talents Cultivation Program of National Administration of Traditional Chinese Medicine (ZYYCXTD-D-202005). This work is also supported by the Scientific Research Funds Project of Liaoning Education Department (JDL2019012). The funders had no role in study design, data collection and analysis, decision to publish, or preparation of the manuscript.

### Grant Disclosures

The following grant information was disclosed by the authors:
Major Scientific and Technological Special Project for "The Drug Innovation Major Project": No. 2018ZX09711001-008.
Innovation Team and Talents Cultivation Program of National Administration of Traditional Chinese Medicine: ZYYCXTD-D-202005.
The Scientific Research Funds Project of Liaoning Education Department: JDL2019012.

## Competing Interests

The authors declare there are no competing interests.

## Author Contributions

- Da-Cheng Hao conceived and designed the experiments, performed the experiments, analyzed the data, prepared figures and/or tables, authored or reviewed drafts of the paper, and approved the final draft.
- Pei Li performed the experiments, analyzed the data, prepared figures and/or tables, authored or reviewed drafts of the paper, and approved the final draft.
- Pei-Gen Xiao conceived and designed the experiments, authored or reviewed drafts of the paper, and approved the final draft.
- Chun-Nian He conceived and designed the experiments, analyzed the data, prepared figures and/or tables, and approved the final draft.

## Field Study Permissions

The following information was supplied relating to field study approvals (i.e., approving body and any reference numbers):

Field experiments were approved by the Research Council of the Institute of Medicinal Plant Development, Chinese Academy of Medical Sciences (project number: 2018ZX09711001-008).

## Data Availability

The data are available at China National GeneBank (CNGB): CNX0280429–0280435, CNX0280421–0280425, CNX0350400–CNX0350404, CNA0036084–CNA0036093, CNA0019224–0019228, CNX0350408 and CNX0350409.

https://db.cngb.org/search/project/CNP0001489/.

## Supplemental Information

Supplemental information for this article can be found online at http://dx.doi.org/10.7717/peerj.12428#supplemental-information.

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
