# Peer review of "Dissection of full-length transcriptome and metabolome of *Dichocarpum* (Ranunculaceae): implications in evolution of specialized metabolism of Ranunculales medicinal plants"

_PeerJ, doi:10.7717/peerj.12428_

## Round 0.1 · original submission · Major Revisions

Both reviewers have brought up many suggestions to improve your manuscript.

Reviewer 1 ·

Basic reporting

The submitted manuscript entitled as "Dissection of full-length transcriptome and metabolome of Dichocarpum (Ranunculaceae): Implications in evolution of specialized metabolism of Ranunculales medicinal plants" by Dr. Hao and colleagues. This manuscript deciphered the full-length transcriptomes and metabolomes of five Dichocarpum species endemic in China through the third-generation high-throughput sequencing and metabolomic techniques. They obtained 71,598 non-redundant full-length transcripts, and twenty-seven orthologs extracted from trancriptome datasets were concatenated to reconstruct the phylogenetic tree and found that the extensive correlations between metabolite content and gene expression.
This work is meaningful, which will promote the insights into the transcriptomic and metabolomic landscapes of Dichocarpum, which will be usefully for further studies on genomics and application of Ranunculales plants.

Experimental design

The authors are also recommended to examine the expression of some representative genes by conduct Quantitative Real-time PCR (qPCR) to validate the transcriptomic results.

Validity of the findings

About transcriptomic and metabolomic analysis, the authors are recommended to give more comparison with previous studies in the results and (or) discussions.

Additional comments

The abstract of this manuscript should be improved to make it more concise. There are a number of grammatical errors in the manuscript and should be proof-read by a native English speaker or a professional language editing service.

Reviewer 2 ·

Basic reporting

The authors compared the transcriptome and metabolome among the five Dichocarpum species, and proposed biosynthetic pathways of BIAs in Dichocarpum.

Experimental design

Major points:
1. A better phylogenic tree with more species is needed to replace Figure 3. How do the five species fit into a previous published the phylogenic tree of the Dichocarpum family? Please refer to: https://www.sciencedirect.com/science/article/abs/pii/S1055790316304493
2. Please clarify the experimental materials – What tissues or organs at what developmental stages were used in the study. Gene expression and metabolism are very different in different tissues/organs at different developmental stages.
3. Please compare the transcriptome and metabolome results with other model organisms, eg. Arabidopsis and Chlamydomonas.
4. What is the significance of the BIA biosynthetic pathway? Are there candidate compounds that have potential or identified medical effects in the BIA biosynthetic pathway?

Validity of the findings

Minor points:
1. The authors stated that CDS length distributions and transcriptomes are similar in the five species. If so, why there are so many unique transcriptome sequences in the five species?
2. Which tissues or organs are known to be enriched with annotated KEGG pathways such as “Phenylpropanoid biosynthesis” and “Glutathione metabolism?”
3. Line 70-73 Lack of citation.
4. Line 84-85 The meaning of the sentence is not clear. What does the “language” mean?
5. Figure 1D. What are the outliner incRNAs? Do those outliers share some similarities in sequences or locations in the genomes?
6. Figure 2A. Why does SCJY have very different interval distribution pattern than the other four species?
7. Figure 2C. Why do gene expression patterns seem to have poor correlations among the five species?
8. Figure 2D. Please make clear annotations on the gene clusters.
9. Figure 4 & Figure 5 – a lot of the information in the figure legends belong to the method section. Please rewrite the legends to be more concise and clear.
10. Line 474 – 498 The hypothesis on the whole genome duplication is a big stretch. Please provide clear explanation with more detailed information.

---

## Round 0.2 · Major Revisions

Both reviewers are satisfied with the progress made in the revised manuscript. However, the Section Editor, Dr. Gerard Lazo, brought up some additional comments on your manuscript as following:

"It is good that the raw read data is available, but it should not be up to the reader to re-create the assemblies based on that data. Samples of the assemblies should be provided to add validity to the findings. I do see that sequence data has been made available for review only, but the annotations created and the characteristics associated with the evolutionary differentials need to be made available for validation.

The manuscript should not move forward if this is not made available. There is very detailed information within the manuscript of file creation, but the files are not available; it basically becomes a story with no grounds to believe in without examples (figures with categories is just that, a picture). I don’t think this can move forward unless some concessions are made to share data pertinent to the purpose of the manuscript.

The sequences should be placed at a public third-party resource such as NCBI, or in the transcript assembly resource. Especially when presenting data in figures (like 1A) you only allow the reader to trust your assessment; there needs to be a platform to test your findings.

Likewise, when characterizing expression data it is generally accepted that annotations regarding tissue, biological, and molecular function can extend the characterization of the transcriptome. Journal manuscripts are often scanned by text-mining software that locates and extracts core data elements, like gene function. Adding standard ontology terms, such as the Gene Ontology (GO, geneontology.org) or others from the OBO foundry (obofoundry.org) can enhance the recognition of your contribution and description. This will also make human curation of literature easier and more accurate. None of this was visible.

The manuscript was an interesting read, but without providing data to validate the findings I would require a more open manuscript. I have also included some markup for suggested revisions; however, the manuscript is still in need of a major proofreading."

Reviewer 1 ·

Basic reporting

The authors clearly clarified the questions that I raised. Basically, I am satisfied with the current version.

Experimental design

The authors clearly clarified the questions that I raised. Basically, I am satisfied with the current version.

Validity of the findings

The authors clearly clarified the questions that I raised. Basically, I am satisfied with the current version.

Additional comments

I have no more comments.

Reviewer 2 ·

Basic reporting

The manuscript was well edited and written. The authors answered all the questions with plenty of details. I support to accept the article.

Experimental design

NA

Validity of the findings

NA

Additional comments

Two minor things:
1. The world “defense” were misspelled as “defence” - line 121 and 132.
2. The color scheme could be improved. The same species could use the same color in all the figures.

---

## Round 0.3 · accepted · Accept

Authors have made progress to address the concerns of the reviewers.